# Generating Data-Driven Reasoning Rubrics for Domain-Adaptive Reward Modeling

## Abstract

An impediment to using Large Language Models (LLMs) for reasoning output verification is that LLMs struggle to reliably identify errors in thinking traces, particularly in long outputs, domains requiring expert knowledge, and problems without verifiable rewards. We propose a data-driven approach to automatically construct highly granular reasoning error taxonomies to enhance LLM-driven error detection on unseen reasoning traces. Our findings indicate that classification approaches that leverage these error taxonomies, or "rubrics", demonstrate strong error identification compared to baseline methods in technical domains like coding, math, and chemical engineering. These rubrics can be used to build stronger LLM-as-judge reward functions for reasoning model training via reinforcement learning. Experimental results show that these rewards have the potential to improve models' task accuracy on difficult domains over models trained by general LLMs-as-judges by +45%, and approach performance of models trained by verifiable rewards while using as little as 20% as many gold labels. Through our approach, we extend the usage of reward rubrics from assessing qualitative model behavior to assessing quantitative model *correctness* on tasks typically learned via RLVR rewards. This extension opens the door for teaching models to solve complex technical problems without a full dataset of gold labels, which are often highly costly to procure.

## 1 Introduction

Using Large Language Models (LLMs) to dynamically self-correct thinking traces is a promising avenue for improving reasoning model performance on complex problems. LLM-as-a-judge verification demonstrates performance benefits at inference time, where it is used to perform rejection sampling and value estimation of trajectories (Lightman et al., 2023b; Gu et al., 2025), as well as at training time, where it can function as an outcome reward model whose feedback informs model fine-tuning (Hosseini et al., 2024). However, many LLMs struggle to recognize errors reliably, especially smaller architectures and those lacking domain-specific customization (**??**). This limitation impacts their viability in training settings and undercuts the perceived data efficiency benefits of LLM judges over other forms of supervised feedback.

Some research suggests that to be effective at most forms of verification, LLMs require either external tools like search engines or large-scale fine-tuning on feedback data (**?**). These interventions are computationally costly and impact the efficiency of training and inference. Furthermore, feedback signals from approaches like preference mining and self-refinement are purely discriminative, which impacts their robustness to new reasoning outputs. Specifically, these signals do not create explicit mechanisms for determining *why* one output is preferable over another, and so it is difficult to distill a true model of the rules underlying the domain to instill in an LLM judge.

We seek a generative form of external feedback that provides the same benefits as these frameworks. We hypothesize that general-domain LLMs are more adept at recognizing task-specific reasoning errors when given explicit error patterns to check against, as opposed to detecting errors with only abstract guidance as to what they may look like. We draw inspiration from inverse-constitutional AI (Bai et al., 2022), which focuses on the problem of *constructing* constitutions, or lists of desiderata, that LLMs should adhere to. We derive constitutions by directly extracting empirical errors made by a model in a given domain, resulting in a granular and representative knowledge store of

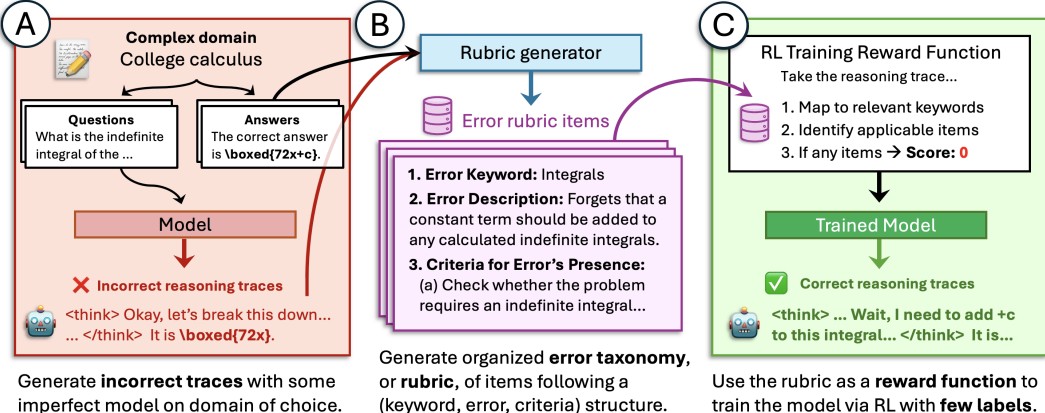

Figure 1: We propose using a knowledge store of errors from inference rollouts to enhance LLM-driven reward functions for model training. We pass incorrect reasoning traces alongside ground-truth answers (Box A) to our "rubric generator" that extracts a set of organized failure modes learned from those incorrect traces (Box B). We then pass this rubric to a LLM classifier that identifies whether a reasoning trace will result in an incorrect answer to serve as a reward function (Box C).

potential errors learned from earlier instances. We refer to this store as a "rubric": a list of error patterns that an LLM should systematically check for in new reasoning traces. We organize our domain-specific error taxonomies as a keyword-indexed text artifact, and use them to inform LLM verifiers during training, illustrated in Figure 1.

We demonstrate that LLM judges equipped with rubrics directly constructed from individual instances of model failure can improve reasoning trace error classification accuracy by up to 11.6% in technical domains (math, coding, etc.) by improving error recall. We then establish that these enhanced judges can effectively train reasoning models in reinforcement learning settings using minimal ground truth labels, approaching downstream validation accuracy of a Qwen3-4B model fine-tuned with verifiable rewards while using less than 20% as many gold labels. Our findings suggest that rubric-enhanced LLM reward functions represent a promising direction for model training without humans in-the-loop or external feedback providing granular error signals.

In summary, our contributions are:

1. An extension of the inverse constitutional AI task to verifiable domains, in which the desired model behavior targeted by the constitution is *correct reasoning processes* leading to *correct answers*.

2. An approach to automatically generate such constitutions, or rubrics, that generalizes to arbitrary technical domains and uses minimal gold labels.

3. Empirical results demonstrating that these automatically generated rubrics improve trace correctness classification as well as downstream model task accuracy when the rubrics are used to generate RL reward signals.

## 2  RELATED WORK

### 2.1  CONSTITUTIONAL AI

Constitutional AI (CAI, Bai et al. (2022)) leverages a list of explicit governing principles (the "constitution") to guide AI model behavior during data curation and preference tuning. This paradigm enables AI systems to proactively identify and correct undesirable outputs through self-critique. Constitutions can be used to synthesize preference data from LLMs and then train a "trait preference model" that scores model responses to encourage or discourage the explicitly stated behavior (Kundu et al., 2023). CAI can teach models safety policies (Guan et al., 2024; Mu et al., 2024) or other general principles (Fränken et al., 2024). Our work can be considered a step towards automatically inducing constitutions of undesirable *reasoning* behaviors from a dataset and then using

the constitution to inform a preference model. Other work has explored methods to discover traits that separate good behavior from bad behavior using a dataset, e.g. VibeCheck (Dunlap et al., 2025), which identifies user-aligned "vibe" statements, while Inverse Constitutional AI (ICAI; Findeis et al. (2025)) and C3AI (Kyrychenko et al., 2025) derive NL principles from annotated preference data. In contrast, our method uses data annotated only with end-task correctness, not preference, in mind.

## 2.2 ERROR TAXONOMIES

A number of papers in topics adjacent to technical reasoning have touched on this notion of "error taxonomies" beyond work in constitutional AI. Many of these papers derive similar ideas from agentic pipelines and robotics, establishing frameworks that are optimized for these domains but can still inspire more "reasoning"-centric work. REFLECT (Liu et al., 2023) collects hierarchical summaries of past experiences to inform failure analysis in robotics. Jain et al. (2022) use SVM boundaries to identify captions in a dataset that "summarize" failure modes of ResNets. Tong et al. (2023) scrape datasets for "erroneous agreement" among outputs of generative multimodal frameworks and generate natural language descriptions of them, Sagar et al. (2024) use scraped errors alongside human feedback to adjust models, and Cornelio & Diab (2024) introduce an online, neuro-symbolic failure identification and recovery framework. Weir et al. (2024) build on the rubric-adjacent argument analysis work of Jansen et al. (2021) to assess the logical consistency of reasoning arguments via rubric-equipped LLMs, and Hashemi et al. (2024) use rubrics to improve human alignment of freeform generated text evaluations. Notably, **?** introduce a method for improving self-correction of complex reasoning by self-asking verification questions based on identified key conditions. Our approach differs from these prior investigations in that it automatically collects and organizes granular error classes for *reasoning tasks* while targeting the complete and detailed coverage of error classes in an individual domain.

## 2.3 REWARD MODELING FOR REASONING MODELS

Recent reward function approaches in reinforcement learning for LLMs and reasoning models include a wide range of methods, spanning from rule-based metrics such as RLVR to learned reward models that can evaluate both process and outcomes of multi-step reasoning (Zhong et al., 2025). Rule-based rewards were critical to the development of "early" reasoning models (Guo et al., 2025), checking for accuracy as well as syntax. However, as rule-based rewards only work for verifiable domains, and may produce false negatives, models trained to assess the correctness of model responses have been adopted for more complex domains (Li et al., 2023; Liu et al., 2025).

Lightman et al. (2023a)'s Verify Step by Step paper demonstrates the benefits of using process supervision, or providing feedback at each reasoning step, for improving LLM alignment on domains like math. Wang et al. (2023) extend this to function without human annotations by using automatically constructed step labels. Luo et al. (2023), Sun et al. (2023), and Yang et al. (2024) propose methods for using LLMs-as-judges for reward modeling during training, and RewardBench was introduced by Lambert et al. (2024) as a benchmark for reward models themselves. Sun et al. (2024) use human-written constitutions as reward models for tasks with non-verifiable rewards, and Cui et al. (2024) produce an LLM feedback dataset, promoting the importance of "scale and diversity". Recently, researchers have begun to adopt rubric-based rewards, but this is so far predominantly constrained to human-generated rubrics (Jia et al., 2025).

## 3 FORMULATING RUBRIC CONSTRUCTION

Our goal in this paper is to develop a system that can autonomously extract reasoning errors from natural language reasoning traces produced by a reasoning model like DeepSeek-R1 (Guo et al., 2025) or Qwen (Yang et al., 2025) over a specific domain and organize them into a rubric. A rubric acts as a checklist where each item represents some catastrophic error, meaning that if a checklist item applies to a reasoning trace, the error in question is highly likely to cause the final answer to be incorrect. We center catastrophic errors to focus on, and hopefully mitigate, the measurable downstream performance impact of reasoning errors. An LLM judge, equipped with the rubric artifact for improved trace classification, takes a trace and runs through the rubric like a checklist, classifying traces with any rubric items "checked" as incorrect. This can then be installed as a

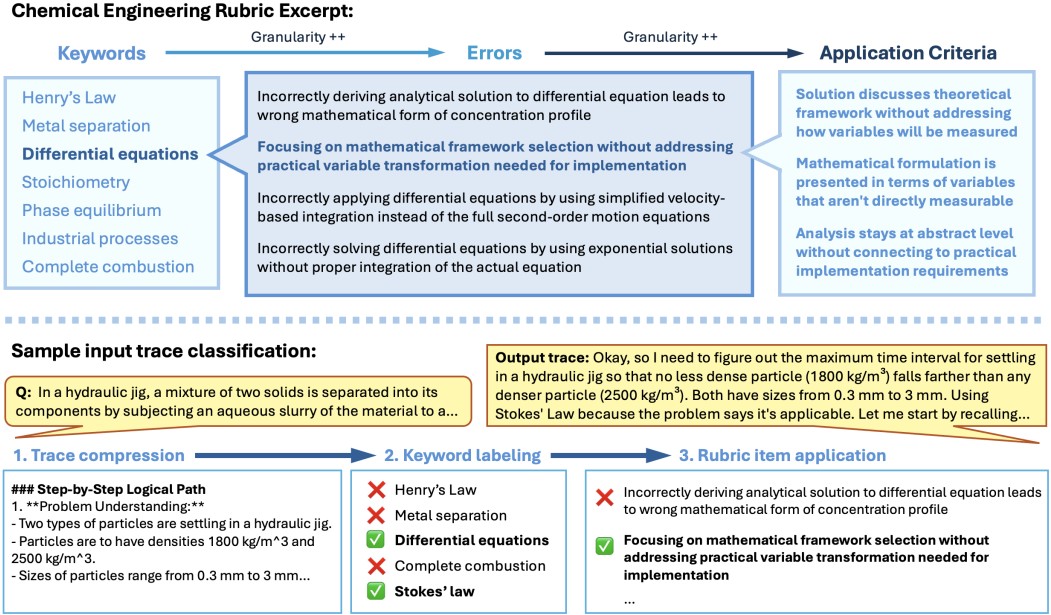

Figure 2: Top: An excerpt from a rubric constructed from chemical engineering problems in the NaturalReasoning dataset, illustrating their hierarchical organization. Bottom: An sample classification made by Claude 3.5 Sonnet using the rubric (showing trace compression, keyword labeling, and rubric item application). The depicted keyword set and rubric items are subsets of the full sets.

reward function that applies this checklist on traces during training for model reinforcement learning, scoring a trace as "correct" only if no rubric items apply to it. We formulate the task of rubric construction below.

**Input**  Our inputs are training trace set $S = \{(q_i, o_i, t_i, y_i)\}_{i=1}^N$ and validation trace set $S' = \{(q'_i, t'_i, y'_i)\}_{i=1}^{N'}$ s.t. $q \in Q \sim D$ and $q' \in Q' \sim D$, where:

- $q \in Q$ are natural language strings that describe the parameters of a reasoning problem and serve as input to a reasoning model $f$.

- Distribution $D$ is some domain of reasoning problems that share some unknown error taxonomy.

- $o \in O$ are correct ground truth solutions to the reasoning problems.

- $t \in T$ are reasoning traces produced by the reasoning model. These are natural language strings that are generated by the model leading up to (and including) its final answer $\hat{o}_i = f(q_i, t_i)$.

- $y_i \in \{0, 1\}$ are binary correctness labels following some scoring function $y_i = g(o_i, \hat{o}_i)$. For example, $g$ could be an LLM providing with a scoring prompt.

**Output**  Our desired output is a failure taxonomy $\Phi$, which parameterizes the predefined *trace classifier* $h_\Phi(t_i) = \hat{y}_i$ (distinct from our evaluation function $g(o_i, \hat{o}_i) = y_i$). $\Phi$ is a set of rubric items $\phi$, or natural language strings describing some behavior that can be observed in a hypothetical trace.

**Hierarchical organization**  As the resulting rubrics can be large, our error items are each labeled with a general keyword that helps an LLM map the correct rubric items to the relevant traces. Then, at inference time, the LLM judge first labels each trace with relevant keywords using the full list of keywords from the taxonomy, and then these labels can be used to filter which rubric items are compared against the traces in a second forward pass. Hierarchy details are elaborated in §4.

# 4 METHOD

In this section, we detail how rubrics are generated. The system can be broken down into two core components: Trace compression (§4.1) and rubric item extraction (§4.2). We describe each component in detail alongside its motivation below. We provide an illustration of a generated rubric and its application in Figure 2.

## 4.1 TRACE COMPRESSION

There is a distinction that needs to be made between the logical path to a solution, and the exploration process a model may undergo to arrive at that final solution. While in theory, the "optimal" reasoning trace is simply the former, in many models the exploration process is a core component enabling it to eventually converge to the correct answer. In some domains, unguided exploration is explicitly necessary to reach a well-formed solution. In open-ended philosophy problems, for example, multiple angles of an idea must be considered to arrive at a satisfactory answer.

The errors that we aim to identify are those that directly and negatively impact the final solution, and so exploration of incorrect approaches that are not incorporated into the final solution should not be included in a rubric. However, this material makes up a large portion of traces outputted by some models, especially those trained primarily via self-supervised methods. Therefore, we first "compress" traces into summaries that outline all of the logical steps taken by the model that influence the eventual final answer. We accomplish this through an LLM call. The downstream impact of this compression is explored more in §C.

## 4.2 EXTRACTING RUBRIC ITEMS

As the intended downstream use case of these generated rubrics is to serve as a resource for LLMs classifying trace correctness, the rubric items must be written such that they can easily be applied to new traces by LLMs. While rubric items must be detailed enough to be applied consistently by a judge model, they must be concise enough to not result in an overly long rubric, which increases the likelihood of application error unnecessarily. To enable rubrics to be long without impacting classification performance or compute, we aim to classify traces via two distinct classification stages: (1) A high-recall, low-granularity classification stage, followed by (2) a high-precision, high-granularity classification stage. This classification approach requires rubric items to comprise three distinct fields:

1. **The main error description:** A simple, straightforward explanation (less than 25 words) describing the error, and possibly a very brief overview of why it can occur.

2. **A keyword:** A word or short phrase that can be used to identify traces that might be susceptible to the error. This should be low granularity and optimized for recall: If this keyword is not relevant to a trace, then the corresponding error cannot be applicable.

3. **Verification details:** One or more descriptive explanations for how a model could tell if the error exists in that argument. If one of these descriptions matches up with the contents of a new reasoning argument, then that means the error has occurred in the trace.

The classification approach using rubrics is illustrated in the bottom half of Figure 2. First, the compressed trace is compared against the full set of keywords, and appropriate keywords are tagged. Then, all rubric items associated with those keywords are presented to a model alongside the compressed trace, and the classifier determines which ones (if any) apply to the trace.[1] If a trace is tagged with any rubric items, our classifier labels it as incorrect.

We extract these rubric items by first passing the incorrect (compressed) traces from the training set to an LLM alongside the problem they attempted to solve and its correct solution, when available. We ask the model to identify potential issues with the compressed trace that would cause it to produce the incorrect answer. We provide an example rubric item to guide the model's output. To further reduce rubric size, we then prompt an LLM to group related keywords, which typically

---

[1]When classifying unseen traces, the classifier is also given the trace the error item was extracted from as an example of an applicable reasoning pattern.

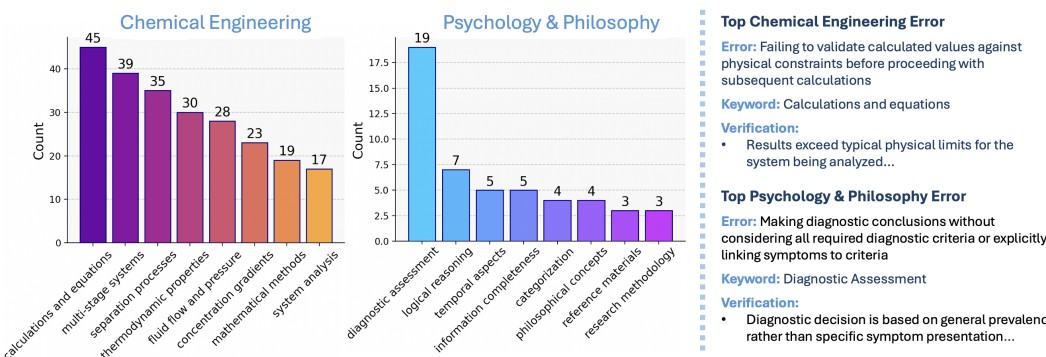

Figure 3: Distribution of most applied rubric item keywords over a technical domain and non-technical (and qualitative) domain, alongside the top most-applied rubric item for each trace set. Both domain problems taken from the Meta NaturalReasoning dataset.

lowers the number of keywords overall by approx. 50%. A comparison of common keywords for technical and non-technical domains are shown in Figure 3.

## 5 EXPERIMENTS

We aim to answer three main questions: **(R1)** Can rubric artifacts improve the specificity and overall accuracy of an LLM trace correctness classifier? **(R2)** Can rubric-augmented classifiers make stronger reward functions than a traditional LLM-as-a-judge reward function? **(R3)** How do rubric-augmented reward functions compare against verifiable reward functions (e.g. string matching) on downstream task accuracy?

We address these questions across two primary experiments: First, we assess how well rubric-augmented LLMs can classify reasoning trace correctness compared to standard LLMs (**R1**), and then we assess how well rubric-augmented LLMs can serve as reward signals during RL training compared to standard LLMs and verifiable reward setups (**R2** and **R3**). Prompts and additional experimental details for these tasks are included in §B, §D, and §E.

For experiments assessing rubric quality as an artifact for trace classification, we consider the following metrics: **Specificity,** or $\frac{TN}{FP+TN}$ (the percentage of incorrect traces classified as incorrect by the LLM), **balanced accuracy,** or $\frac{TN}{2(FP+TN)} + \frac{TP}{2(TN+TP)}$ (the average of the specificity and recall), and **F0.5,** $(1 + \beta^2)\frac{PR}{\beta^2 P+R}$ where $\beta = 0.5$, $P = \frac{TP}{TP+FP}$, and $R$ is recall. This is the balanced precision and recall with additional weight on precision.

### 5.1 DATASETS

Our primary data constraints are that the domains should be complex enough that they elicit sufficiently long traces, and that the answers to the problems should be verifiable in a relatively consistent and meaningful way. We explore classification behavior on more qualitative data in §B.5, but for main experiments, we select the following datasets:

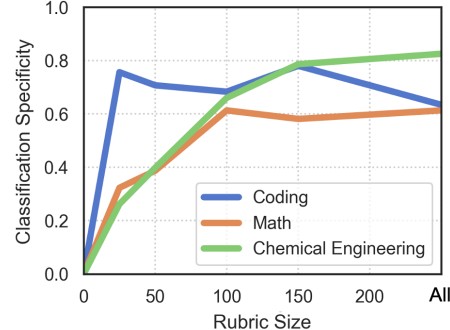

Figure 4: Experiment 1 ablation: # of rubric items vs. classification specificity.

**SWE-Bench** (Jimenez et al., 2024) tests our approach in the domain of coding and software development. SWE-Bench is a collection of issue-pull request pairs from GitHub repositories that evaluates Python code patches via a set of unit tests. **NuminaMath** (Li et al., 2024) tests complex mathematical reasoning. NuminaMath is a collection of high school/competition-level math problems paired with chain-of-thought style ground truth solutions collected via PDF scanning. We filter the dataset for problems labeled as calculus, geometry, and number theory. Finally, we extract a subset of chemical engineering

problems from Meta's **NaturalReasoning** (Yuan et al., 2025) dataset for more open-ended problem-solution pairs. NaturalReasoning is a broad collection of problems from various disciplines ranging from social sciences, coding, math, physical sciences, and humanities.

## 5.2 EXPERIMENT 1: CLASSIFYING REASONING ERRORS

We evaluate the performance of a rubric-augmented LLM on identifying traces that lead to incorrect answers. This experiment targets **R1**, or whether rubric artifacts can improve the *specificity and accuracy* of an LLM-based trace correctness classifier.

**Data**   We collect long-form reasoning traces for each dataset listed in the previous section. SWE-Bench traces are generated with a tool-calling variant of Claude 3.5 Haiku [2], and NuminaMath and NaturalReasoning traces are generated with DeepSeek-R1 (Guo et al., 2025), sourced from existing HuggingFace resource datasets [3] [4]. For NuminaMath and NaturalReasoning, R1-generated final answers are scored as correct or incorrect by a Claude 3.5 Sonnet model that is provided access to the question, R1 answer, and ground truth solution. This prompt is designed to frame the scoring task as a homework grading scenario. For each domain, we sample 450 to 800 problems, depending on data availability, and split 80/20 between train and validation.

**Methods**   For each domain, a rubric is constructed using the incorrect traces from the training question-trace pairs from each dataset with Claude 3.5 Sonnet v2, resulting in rubrics of approximately 250 items each. Keywords are illustrated in Figure 3. Keywords are not clustered for NuminaMath due to the narrow range of problem types, keyword clustering influence on performance is explored in §C. We compare a rubric-augmented LLM classifier against a standard LLM (using a prompt with no rubric). For LLM calls, we use Claude 3.5 Sonnet v2[5].

The baseline LLM classifier judges traces using a single prompt[6], while the rubric-augmented classifier use a two-stage classification pipeline. In the first pass, the classifier identifies which keywords generated alongside rubric items map to each validation trace, and then in the second pass, all rubric items associated with the mapped keywords are compiled into a mini-rubric that is compared against the reasoning trace. If any of the rubric items are tagged by the classifier as applying to the trace, the trace is marked as incorrect. If no items are applied, the trace is marked as correct.

**Results**   Experimental results are included in Figure 5. The results illustrate that the rubric augmentation can be highly effective at improving specificity, or recognizing when an incorrect trace is incorrect. The rubric-augmented classifier also consistently results in higher overall balanced accuracy when compared against the baseline classifier, although this margin is generally smaller. These results clearly illustrate the strong tradeoff between specificity and recall, or the ability to correctly classify correct traces. Errors in reasoning traces do not always negatively impact the downstream final answer, as the relationships between individual model steps are often tenuous and complex Levy et al. (2025). We believe that jointly optimizing specificity and recall further in trace classification is an exciting direction for future research.

## 5.3 EXPERIMENT 2: RUBRIC REWARD FUNCTIONS

**Overview**   Current approaches to reward modeling for reinforcement learning with language models often rely on simple evaluation metrics that focus primarily on the correctness of final answers against ground truth solutions, requiring a large set of verifiable ground truth solutions or compute-intensive answer verification systems. By enhancing the capacity of LLM judges to reliably evaluate reasoning traces, we can develop more effective reward functions that optimize not just for correct answers but for sound reasoning processes using a small set of ground truth solutions. In this section, we explore how our rubric-augmented trace classifiers can be used to improve RL training in

---

[2]Link to SWE-Bench submission link

[3]Link to NuminaMath source

[4]NaturalReasoning source

[5]Link to Claude 3.5 model card

[6]Some SWE-Bench traces were longer than the model's context size. The prefixes of these traces were cut s.t. they could fit within the model's context window.

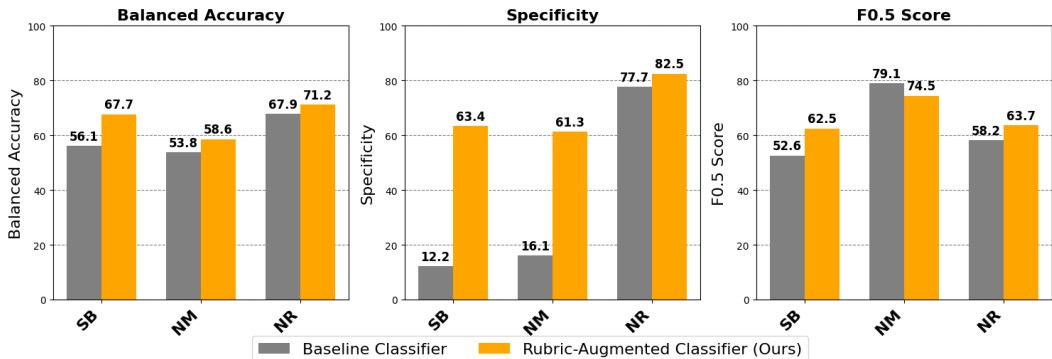

Figure 5: Rubric-augmented Claude 3.5 Sonnet outperforms standard prompted Sonnet at identifying incorrect reasoning traces across three diverse domains: SWE-Bench (SB), NuminaMath (NM), and chemical engineering problems from NaturalReasoning (NR). We present balanced accuracy (mean accuracy with correct and incorrect traces weighted equally), specificity (# of incorrect traces classified as incorrect), and F0.5 (measuring precision and recall with additional precision weight).

complex domains where ground truth labels are scarce. This experiment targets **R2** and **R3**: We assess whether a rubric-augmented classifier's ability to act as a reward function for RL training can improve downstream task accuracy over a standard LLM or verifiable rewards.

**Data** We perform this experiment in the math and coding domains, as they enable explicit verifiable rewards for a completely grounded comparison (independent of LLMs) against the new LLM-based reward approach. For each of these datasets, we first fine-tune a Qwen3-4B model (thinking mode) (Yang et al., 2025) using SFT with a subset of 1.6K samples of question-ground truth solution pairs, using a cross-entropy loss and a learning rate of 1e-6. We fine-tune for only one epoch as convergence occurs quickly for these domains due to limited drift from the original pretraining data. For NuminaMath, we partition the remaining dataset on problem type and use it to randomly sample a second subset of 1.4K training problems and 340 validation problems to serve as our (disjoint) reinforcement learning dataset. For SWE-Bench, we sample 1.4K problems from the train dataset for training and 80 problems from the test set for validation.[7]

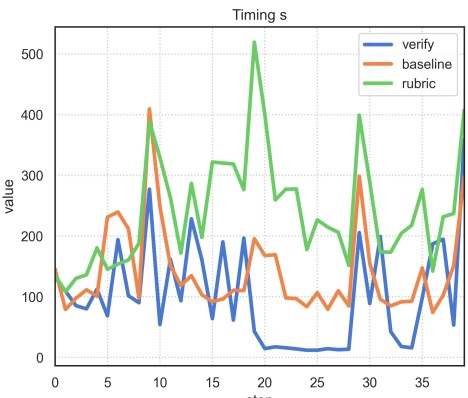

Figure 6: Time (seconds) per step in Experiment 2 across reward methods. This highlights that the computational cost of the rubric method is not significant compared to the baseline classifier (but, naturally, is still higher than some methods that rely on verifiable rewards).

**Training Setup** We train Qwen3-4B using the DAPO RL algorithm (Yu et al., 2025). For DAPO, the model was configured with a maximum response length of 5120 tokens for math and 10000 tokens for coding. For our advantage estimation, we employed GRPO. Models are trained on 8 40GB Nvidia A100s. For our validation reward on NuminaMath we used MathVerify package [8].

**Methods** As comparison models, we include Claude 3.7 Sonnet, Claude 3.5 Haiku, DeepSeek-R1, and Mistral-7B[9]. We again cut off math responses at 5k tokens and coding responses at 10k. For each domain, we use the fine-tuned Qwen3-4B model (post-SFT, pre-DAPO) to generate traces for a rubric generation dataset. For math we generate traces using a sample of our RL training problems

---

[7]Data limited to shorter inputs (less than 25000 characters for RL and 35000 characters for rubric construction for memory and compute efficiency).

[8]https://github.com/huggingface/Math-Verify

[9]Link to Mistral model card. Mistral traces are cut off at 5k for both domains.

| Benchmark | Baseline Models | | | | RL w/ GT | RL w/o GT | | |
|---|---|---|---|---|---|---|---|---|
| | **Sonnet** | **R1** | **Haiku** | **Mistral** | **VR** | **No RL** | **BL** | **Rubric** |
| Model Size[10] | $\sim$100B+ | 37B | $\sim$20B+ | 7B | 4B | 4B | 4B | 4B (Ours) |
| NM (Acc) | 21.7 | 15.5 | 16.3 | 3.7 | 16.1 | 5.5 | 10.5 | **15.2** |
| SB (Acc) | 28.8 | 8.8 | 1.3 | 0.0 | n/a | 0.0 | 0.0 | **2.5** |
| SB (Comp) | 77.5 | 30.0 | 2.5 | 15.0 | n/a | 0.0 | 2.5 | **20.0** |

Table 1: Validation accuracy and patch completion results post-RL training demonstrate that the rubric-augmented LLM-based reward function produces models with higher output quality than the baseline LLM reward function (using a prompt with no error rubric) on both domains, as well as the standard verifiable rewards approach on the math domain. The rubric-augmented reward function performs best when paired with a penalization term that discourages overly short reasoning traces.

set and for SWE-Bench use another subset of the original dataset's test data (as these problems are verifiable, unlike the train set) and verify them with ground truth answers to produce a set of 200 verifiably incorrect traces. Rubrics are generated with these incorrect trace sets using Claude 3.5 Sonnet v2, following the setup detailed in the previous experiment. Due to the brevity of the produced traces post-fine-tuning, we omit the trace compression step before extracting rubric items. We compare three different training reward functions:

**Gold Match** is identical to the validation reward function. Model responses are taken as-is and compared against a ground-truth answer using MathVerify. SWE-Bench version is not included due to high resource costs of running the benchmark unit tests in an RL setting. **Baseline Classifier** replicates the baseline classifier used in Experiment 1 using Amazon Nova Lite as the LLM judge.[11] Provided with the question and model response, the LLM assesses whether or not the answer is likely to be correct or not, outputting binary labels. **Rubric-Augmented Classifier** replicates the rubric classifier used in the classification experiments (§5.2), also using Amazon Nova Lite with a rubric constructed by Claude 3.5 Sonnet, detailed above and also using binary $[0, 1]$ integer labels depending on the trace correctness classification. As with the rubric construction, we omit the trace compression step at inference time.

**Results** Results are included in Table 1. Timing information is shown in Figure 6. For both datasets we report validation accuracy, and for SWE-Bench we also report completed patch rate (the number of not-empty patches that do not cause an execution error) due to task difficulty and low performance on final patch correctness of most of the tested models. We report the highest performance achieved during training (computed every ten steps over the full validation set for each domain). The higher performance of the rubric-augmented LLM reward on both domains indicates that it is a strong alternative to traditional training methods while requiring significantly

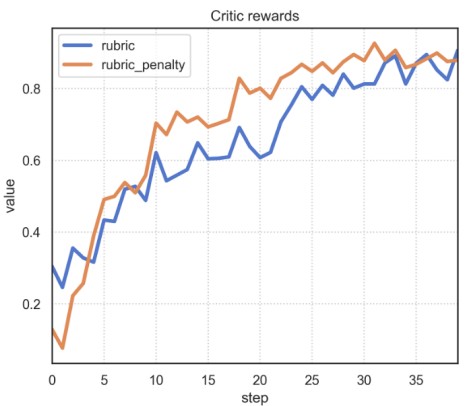

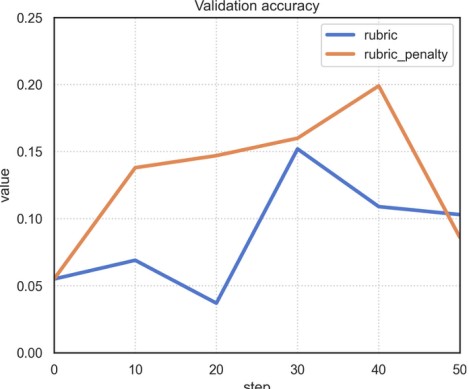

Figure 7: Train/validation for rubric rewards with and without a trace length penalty.

fewer ground truth labels: The rubric method approaches the verifiable rewards setup for Numina-Math and improves patch completion percentage by a full 17.5% over the baseline for SWE-Bench,

[10]Claude model parameter counts are reported as a rough, estimated lower bound, sourced here. We report the number of activated paramers at runtime, so report 37B for R1 instead of 671B.

[11]Link to Amazon Nova model.

notably outperforming the larger Claude 3.5 Haiku model on both metrics. We hypothesize that further investigation into leveraging rubrics as an artifact to improve reward functions can yield greater performance improvements, such as incorporating other qualitative feedback signals, indicated by the improvement achieved by adding a trace length penalty in Figure 7.

## 6 CONCLUSION

This work explore methods to automatically create domain-specific reasoning error taxonomies, significantly enhancing the capability of LLMs to identify their own errors and addressing a fundamental challenge in employing LLM-as-judge frameworks. This methodology reduces the need for extensive manual curation of gold labels while accounting for the unique reasoning patterns and potential pitfalls characteristic of each domain.

Our work enables efforts in training reasoning models in domains where ground truth data is ambiguous or expensive to obtain. By providing a structured framework for error identification requiring only a few gold labels, we address a critical bottleneck in the development of LLMs for specialized knowledge domains. The potential for these rubric-enhanced judges to support human-AI collaborative reasoning is a line of exciting future work enabled by this research.

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

## A  ADDITIONAL BASELINE EXPERIMENTS

We evaluated a range of baseline prompts to compare against in our experiments detailed in §5.2 and §5.3. In Table 2, we report our primary classification metrics for a representative set of 6 baseline approaches. These approaches are:

1. **Baseline**: The baseline used in the main experiments, using the prompt listed in §E.11 and §E.14 alongside the full model reasoning trace. The classifier is simply asked whether, given the model output and the original question, the full reasoning trace results in a correct answer or not.

2. **Baseline Alternate 1**: The same setup as Baseline, but only the first 75% of the trace is presented to the classifier (determined by line count). The motivation behind this is that, empirically, the baseline classifier is highly likely to score a trace as correct if it appears to have converged on a final answer, regardless of answer soundness. By omitting the last decisions made by the model in its thought process, the overall error specificity is likely to increase.

3. **Baseline Alternate 2**: Uses the prompt listed in §E.12 and §E.15. Following the logic detailed for Alternate 1, we provide the full trace, but tell the classifier model that it is only a snippet of the full reasoning passage.

4. **Baseline Alternate 3**: The same setup as Alternate 2, but the last 75% of the trace is again cut off as in Alternate 1.

5. **Baseline Alternate 4**: Uses the prompt listed in §E.13 and §E.16. The prompt indicates that only a portion of the reasoning trace is provided, but instead of asking if the trace will result in a correct output, asks whether the model should continue thinking or provide an answer immediately. The full trace is provided.

6. **Baseline Alternate 5**: The same setup as Alternate 4, but the trace is cut off in the manner described in Alternate 3 and Alternate 1.

| | SWE-Bench | | | NuminaMath | | | NaturalReasoning | | |
| --- | --- | --- | --- | --- | --- | --- | --- | --- | --- |
| **Domain** | **BA** | **S** | **F0.5** | **BA** | **S** | **F0.5** | **BA** | **S** | **F0.5** |
| Rubric | .677 | .634 | .625 | .586 | .613 | .745 | .712 | .825 | .637 |
| Baseline | **.561** | .122 | **.526** | .538 | .161 | **.791** | .679 | .777 | .582 |
| BL Alt. 1 | .497 | .463 | .452 | .538 | .258 | .777 | .674 | **.893** | **.628** |
| BL Alt. 2 | **.561** | .122 | **.526** | .527 | .129 | .789 | **.697** | .777 | .601 |
| BL Alt. 3 | .442 | .415 | .399 | .533 | .216 | .780 | .689 | .787 | .596 |
| BL Alt. 4 | .537 | .073 | .513 | **.548** | .290 | .780 | .598 | .524 | .464 |
| BL Alt. 5 | .444 | **.732** | .260 | .511 | **.355** | .736 | .546 | .728 | .405 |

Table 2: Alternate baselines compared against the rubric approach and the primary baseline used in §E.12 and §E.15. All experiments are run with Claude 3.5 Sonnet. BA = Balanced Accuracy and S = Specificity.

Generally, the original baseline method achieves a strong balance between balanced accuracy and F0.5 score across the three domains compared to the other methods. As shown in the results, there is a clear tradeoff between the different primary metrics. As hypothesized, only providing the first 75% of the trace increases specificity, but generally negatively impacts balanced accuracy and F0.5. However, none of the baseline approaches outperform the rubric method on more than one of the three metrics for any domain, indicating that the rubric approach is generally superior in terms of error identification for technical content.

Additionally, we re-ran the classification experiment using an open-source model, Qwen3-235b, for both rubric generation and classification. We report these results alongside the Claude 3.5 Sonnet results in Table 3.

|  | **SWE-Bench** | | | **NuminaMath** | | | **NaturalReasoning** | | |
| --- | --- | --- | --- | --- | --- | --- | --- | --- | --- |
| **Domain** | **BA** | **S** | **F0.5** | **BA** | **S** | **F0.5** | **BA** | **S** | **F0.5** |
| Claude Baseline | .561 | .122 | .526 | .538 | .161 | **.791** | .679 | .777 | .582 |
| Claude Rubric | **.677** | **.634** | **.625** | .586 | **.613** | .745 | **.712** | **.825** | **.637** |
| Qwen Baseline | **.634** | .268 | **.571** | .526 | .108 | **.791** | .600 | .272 | .456 |
| Qwen Rubric | .536 | **.854** | .417 | **.544** | **.231** | .786 | **.674** | **.476** | **.518** |

Table 3: Open-source classification results.

## B  RUBRIC ANALYSIS

### B.1  KEYWORD VISUALIZATION

Below in Figure 8 we provide histograms of the top 20 most common keywords (in terms of # of rubric items, as opposed to in terms of the number of *applied* rubric items over the validation set of traces as in Figure 3) for the rubrics produced in §5.2.

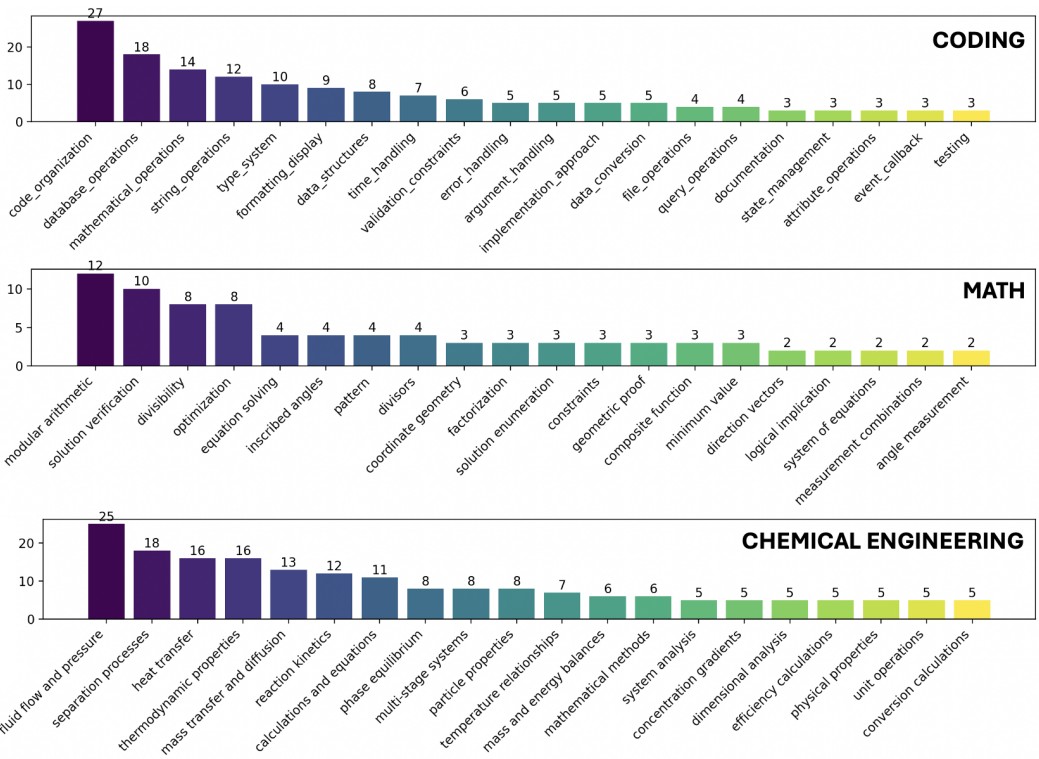

Figure 8: Frequency of top twenty keywords from generated rubrics on target technical domains, in terms of the total number of rubric items.

### B.2  RUBRIC STATISTICS

Below, we report the total number of training and test traces for the datasets described in §5.2, the corresponding rubric size, and the number of keywords (after clustering for chemical engineering and coding).

| Domain | Train set | Val. set | Rubric size | # Keywords |
|---|---|---|---|---|
| **Chemical Eng.** | 636 | 158 | 296 | 90 |
| **Math** | 476 | 124 | 250 | 169 |
| **Coding** | 406 | 73 | 234 | 89 |

Table 4: Dataset and corresponding rubric statistics for the §5.2 classification experiment setup.

### B.3 CLASSIFICATION CONFUSION MATRICES

In Figure 9, we show the confusion matrices for the rubric-augmented classifier's outputs on the validation set for the experiment described in §5.2.

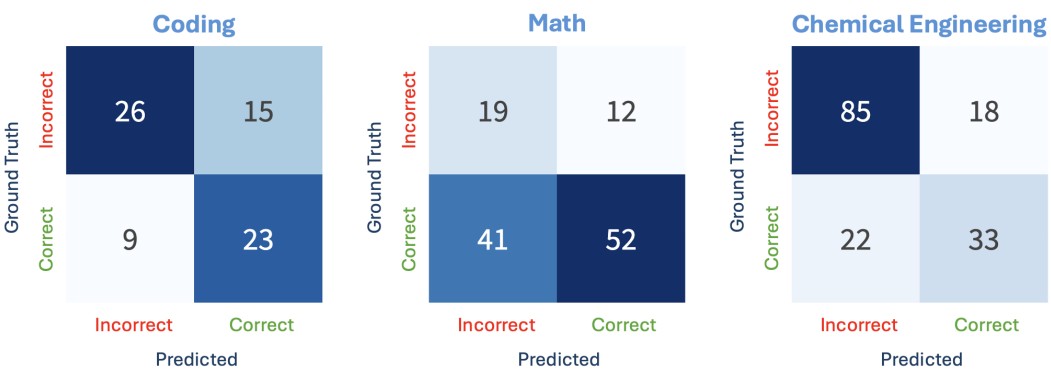

Figure 9: Confusion matrices corresponding to the experimental results illustrated in Figure 5. As shown, false negatives tend to be more common among the rubric approach than false positives, given the total number of positives and negatives per validation set.

### B.4 TRAINING SET CLASSIFICATION ACCURACY

Below in Table 5 we report the primary classification metrics over the training trace set used to construct the rubrics used for classification over the validation set in §5.2.

| | SWE-Bench | | | NuminaMath | | | NaturalReasoning (Chem) | | |
|---|---|---|---|---|---|---|---|---|---|
| **Domain** | **Bal. Acc.** | **Spec.** | **F0.5** | **Bal. Acc.** | **Spec.** | **F0.5** | **Bal. Acc.** | **Spec.** | **F0.5** |
| **Baseline** | .562 | .123 | .509 | .515 | .098 | .524 | .662 | .676 | .687 |
| **Rubric** | .669 | .438 | .586 | .566 | .431 | .545 | .703 | .662 | .722 |

Table 5: Baseline Classifier vs. Rubric-Augmented Classifier results when evaluated on the training set used to construct the rubrics in §5.2.

As illustrated above, the results achieved on the training set are not notably better than those achieved over the validation set as reported in Figure 5. This indicates that the approach is highly generalizable and robust to unseen traces, but simultaneously suggests that the classification approach outlined in §4.2 is not necessarily optimal for aligning erroneous traces with appropriate rubric items. All of the errors constructed for the rubric were mined specifically from incorrect traces in the training set, and so the low specificity values of $\leq .662$ indicate that the classifier is unable to re-identify these errors without the ground truth answers being provided for guidance. This highlights an interesting area for future work that may improve the quality of the rubric approach, and its consequent downstream performance on classification and reward modeling.

### B.5 NON-TECHNICAL RESULTS

We select a sample of $\sim 600$ questions from NaturalReasoning that concern philosophy and psychology questions, set aside a test set of approx. 150 questions, and generate a rubric on the remaining trace set. We evaluate a rubric-based classifier and the baseline approach on the test set, documented in the table below alongside the corresponding confusion matrix in Table B.4.

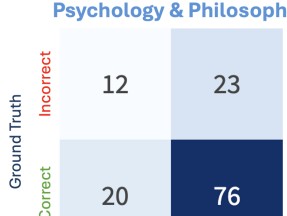

Our hypothesis as to why the method's performance is lower than on the technical domains is not due to inherent drift towards more qualitative failure modes, but due to the dataset construction itself: The joint collection of psychology and philosophy problems (1) encompassed two highly distinct failure mode sets, and (2) has no simple way of obtaining ground truth trace correctness scores without expert trace annotation. Both of these factors make the problem inherently more challenging, and point to tackling them as exciting areas for future work.

| Method | Balanced Acc | Specificity | F0.5 |
|---|---|---|---|
| Baseline | .571 | .225 | .821 |
| Rubric | .567 | .343 | .772 |

# C ABLATION EXPERIMENTS

We consider different alternate rubric construction and classification setups for the three domains considered in §5.2:

- No trace compression for trace classification.
- No keyword clustering.
- A second rubric item classification filter (uses the same prompt as the first rubric item classification, shown in §E.5 and §E.10). This in effect requires the LLM trace classifier to confirm once again that the rubric items applied by the classifier in the previous step are correctly applied, in an attempt to reduce false negatives during classification.

We report the exact numbers below in Table 6. We also consider the impact of different rubric sizes on classification efficacy: 25 items, 50 items, 100 items, 150 items, and the full rubric, the results of which are shown in brief in Figure 4.

| | SWE-Bench | | | NuminaMath | | | NaturalReasoning | | |
|---|---|---|---|---|---|---|---|---|---|
| **Domain** | BA | S | F0.5 | BA | S | F0.5 | BA | S | F0.5 |
| **Baseline** | .561 | .122 | .526 | .538 | .161 | **.791** | .679 | .777 | .582 |
| **Rubric** | **.677** | .634 | **.625** | **.586** | **.613** | .745 | **.712** | **.825** | **.637** |
| **No compression** | .491 | **.951** | .114 | .527 | .323 | .757 | .601 | .602 | .470 |
| **No clustering** | .581 | .537 | .532 | **.586** | **.613** | .745 | .702 | .767 | .601 |
| **Second filter** | .601 | .390 | .551 | .457 | .129 | .740 | .657 | .495 | .508 |
| **Rubric Size** | BA | S | F0.5 | BA | S | F0.5 | BA | S | F0.5 |
| **25 items** | .628 | .756 | .588 | .468 | .323 | .704 | .567 | .262 | .436 |
| **50 items** | .604 | .707 | .556 | .409 | .387 | .608 | .572 | .398 | .439 |
| **100 items** | .560 | .683 | .500 | .554 | **.613** | .708 | .657 | .660 | .531 |
| **150 items** | .609 | **.780** | .565 | .495 | .581 | .640 | .657 | .786 | .560 |
| **All items** | **.677** | .634 | **.625** | **.586** | **.613** | .745 | **.712** | **.825** | **.637** |

Table 6: Ablation experiments corresponding to the experiment setup in §5.2.

The performance drop from removing the trace compression at test time indicates that the unavoidable information loss of this step is less critical than the importance of distilling the trace to a smaller size. In implementation, the resulting prompts were sometimes longer than the context window of Claude 3.5 Sonnet, and so the prompts had to be cut off to obtain classifications. For the domains with more failure modes (not the NuminaMath subset), removing the keyword clustering similarly hurt overall performance.

Somewhat surprisingly, the second filter did not improve F0.5 scores over the normal rubric approach, despite the filtering method being designed to improve recall and precision of the classifier. This result may be related to that of §B.4, in that the classifier itself is likely leveraging the rubric artifact sub-optimally which may be introducing noise.

The math and chemical engineering rubric size trends are not particularly surprising: There is a general positive correlation between increased rubric size and metric performance across the board, although the results demonstrate slightly higher balanced accuracy and F0.5 scores for math at n=25 over respectively larger (but still small, overall) rubrics. The coding results are an outlier, showing higher scores for all three metrics for the smallest rubric size over most other rubrics except for n=150 and the full-sized rubric. This indicates that there are a small number of rubric items that may account for a large portion of the failure modes in the training and validation sets for this domain. Exploring the relative importance of different rubric items, as well as determining the optimal rubric size for a given domain, depending on the domain's breadth and potential for a variety of failure types, is interesting future work. Learning to leverage this information could produce more intelligent and effective classifiers overall.

# D    ADDITIONAL RL TRAINING FIGURES

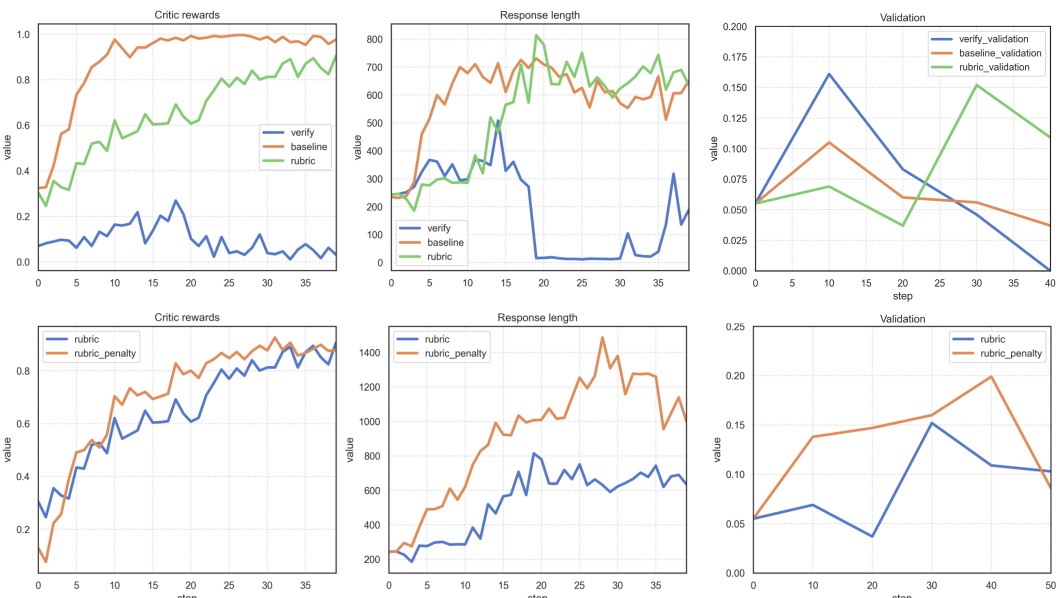

Figure 10: Mean critic reward, mean response length, and mean validation for DAPO training on NuminaMath. Top row shows the training run using the verifiable rewards reward function (Math-Verify), the baseline classifier, and the rubric-augmented classifier, and the bottom row shows the rubric-augmented classifier with and without a length penalty applied to the model output.

As illustrated above in Figure 10, the model is able to learn the LLM-as-judge reward functions much more easily than the verifiable rewards feedback signal, despite selecting the training hyper-parameters based on performance using the latter reward function. This suboptimal behavior is possibly partially explained by the response length, which drops dramatically around when the model achieves its best performance on the training data. The model learns the baseline reward function most quickly, and then seems to proceed to overfit for that signal, resulting in a poor translation to downstream validation set performance.

The second row (partially reported in Figure 7) demonstrates the potential of "hybrid rewards" that incorporate the rubric-augmented classifier into a more complex reward function. Adding a length penalty, $\max(0, \frac{1}{200}(200 - \xi))$, to the reward results in a more stable validation curve that better resembles the critic reward, and improves the best validation accuracy achieved by 4.7% and outperforms the verifiable rewards setup by 3.8%.

Seconds-per-step timing information is included in Figure 6, which shows that the rubric-augmented classifier does not incur a significant increase in runtime compared to the other two reward function approaches.

# E PROMPTS

## E.1 INITIAL TRACE COMPRESSION (GENERAL)

You are given a reasoning trace output by a model trying to answer a question. The reasoning trace involves the model's exploration process to find the logical path to the correct answer. Your job is to distill out the entire logical path that it ultimately converged to and describe it to me in detailed, enumerated steps.

Potential errors should not be omitted from your logical path; provide the final deductive steps as faithfully to the model's trace as possible. The final trace should resemble the deductive steps that a college student would write on a exam to "show their work" to their professor (ideally in complete sentences).

Write your distillation of the model's logical reasoning path, and nothing else, i.e. don't include "here is the distilled path" or anything like that.

TRACE:
"""

{t}
"""

FINAL DISTILLED LOGICAL PATH:

## E.2 RUBRIC EXTRACTION (GENERAL)

You are a reasoning analysis system that (1) identifies errors in students' logical arguments and (2) explains how others can identify these same errors in other arguments.

You will be given a question, correct answer, and the reasoning behind a student's incorrect answer. Identify the error made by the student that caused them to arrive at an incorrect answer. An error description should be concise (max 25 words), but descriptive enough that it could be easily identified in other students' responses in the future. Do not mention the student in an error: Your description should describe a reasoning mistake in the abstract that can be used by other students when solving similar problems, without calling out this specific student. For similar reasons, do not make the error overly specific to the problem, as the error should be sufficiently generalizable to other similar problems that may have different details. Anyone who makes the error on future problems will likely lead to a wrong answer.

You will write instructions in a specific format that will describe how someone could identify the error if it appeared in a different logical argument for a different question. Your instructions should be organized into two parts: A "key concept" and "verification details".

* KEY CONCEPT: This is a short keyword or concept that someone can quickly identify as being present or not present in a reasoning argument to determine if the error could possibly occur in that argument. This should be the lowest level of granularity: If this keyword or concept is present, then the error cannot possibly be present in the argument. Key concepts should be general: You want to aim for recall, not precision.

* VERIFICATION DETAILS: Provided that one or more of the key concepts are present in an argument, write one or more descriptive explanations for how someone could tell if the error exists in that argument. If one of these descriptions matches up with the contents of a new reasoning argument, then that means the error has occurred there.

Write your answers in JSON format with your error as a key and your instructions in another dictionary as the corresponding value, i.e. {{⟨error_description⟩: {{"key_concept": ⟨scanning_keyword⟩, "verification": ⟨verification_details⟩}}}}. Here is an example output:

{{"Failing to verify that atomic/molecular balances are maintained in chemical equations results in a stoichiometric balance violation": {{"key_concept": "chemical equations", "verification": ["Number of atoms of each element isn't equal on both sides of reactions", "Molar ratios in calculations

don't match stoichiometric coefficients", "Product quantities exceed theoretical maximum based on reactant stoichiometry", "Component balances don't account for stoichiometric relationships"]}}}}

Now, here is the data you should use to formulate your output:

Question: "{tq}"

Correct answer: "{ra}"

Student reasoning: "{tt}"

Output the error JSON and ***nothing else***:

### E.3   KEYWORD AGGREGATOR (GENERAL)

A model is trying to match pre-written feedback to different samples of students' work. Here are some examples of the pre-written feedback in the corpus:

{samples}

However, there are more possible feedback pieces than can fit in a single prompt. So, the model first labels each student's work with one or more "key concepts", which it then uses to figure out which feedback might be applicable and should be included in the final prompt.

The following candidate "key concept" labels have been generated. Your job is to consolidate highly similar labels that describe the same concepts, to produce a shorter final list of key concepts.

{keywords}

Group the concepts by the underlying sort of material they describe, and then for each group write a new key concept in the same style that can replace the labels in that group. Organize your output in JSON format with your new labels as keys and the lists of original label indices as the values, e.g., {{"description 1": [1, 15, 62], "description 2": [4, 5, 22, 23, 45], ...}}. Output the JSON and nothing else.

### E.4   RUBRIC KEYWORD APPLICATOR (GENERAL)

You are a reasoning analysis system that identifies core features of reasoning traces produced by other models. You are an expert at tagging these reasoning traces with labels to accurately cover all relevant aspects of the trace so the trace can be searched for in a database later. Given a reasoning trace and list of concept labels, identify each label in the list that describes a concept or idea present in the reasoning trace. You are aiming for complete coverage - every relevant label should be included.

Go down the list item-by-item to see if each one applies to the trace. Write the name of each label that relates to the reasoning trace in a Python list, e.g., ["label1", "label2", "label3", ...]. Output your Python list and nothing else.

All concept labels:

"""

{r}

"""

Question:

"""

{q}

"""

Reasoning trace:

"""

{t}

"""

Matching keywords and concepts (in Python list form):

### E.5 RUBRIC ITEM APPLICATOR (GENERAL)

You are a teacher's assistant. You grade student responses to questions based on whether or not they will arrive at the correct answer and why. For this class, the quality of a student's reasoning generally doesn't matter, and they are only graded on final responses. Due to a server failure, one student's final answer was deleted from the database, leaving only their work leading up to their answer. Your job is to use the student's work to best identify whether the student's final answer was correct or incorrect.

You have been given the teacher's grading rubric that identifies specific reasons why a student's answer may be incorrect based on their reasoning process, and provides specific ways to tell if one of these errors has occurred in a student's work. Use this rubric to predict whether the student's answer should be marked "correct" or "incorrect".

First, read through the student's work. Then, go through the teacher's grading rubric, line-by-line, and see if any of the entries and their descriptions describe behavior in the student's work that will cause them to arrive at the wrong answer. To provide transparent documentation of your own reasoning, write an explanation for each rubric item describing why or why not the item explains why the student got the answer correct or not.

NOTE: The teacher's rubric are suggestions to check for, but just because the student exhibits the erroneous behavior described in a given rubric item DOESN'T mean that they will definitely arrive at the wrong answer. ONLY report back rubric items if you are confident that the erroneous behavior is a highly likely cause for why the student's reasoning arrived at the wrong answer.

Do not skip any items - provide an explanation for each one. Explain one error type per line, and do not abridge the process by saying things like [Continuing through all list items...]. I want all of the items listed, you do not have to ask me again. I do not care how long the output is. Write your reasons in JSON format, e.g., {{1: "explanation for error 1", 2: "explanation for error 2", ...}}.

Then, at the end after your JSON of explanations, you will enter a final grading database entry using ⟨ANS⟩ brackets. Write a final response of "N/A" if you are confident the student ended up at a correct answer. Write the error number(s) of rubric reasons if the student's work ends up at an incorrect answer FOR THOSE SPECIFIC REASONS. Write your answer in ⟨ANS⟩ brackets, e.g. ⟨ANS⟩3, 41⟨/ANS⟩, ⟨ANS⟩22⟨/ANS⟩, ⟨ANS⟩N/A⟨/ANS⟩, etc.

Error rubric: {rubric}

Question: {question}

Student's work: {trace}

Responses:

### E.6 INITIAL TRACE COMPRESSION (CODING)

You are given a reasoning trace output by a model trying to solve a PR request. The reasoning trace involves the model's exploration process to find the correct solution. A critic model will be used to determine whether any mistakes were made in the process, but the existing trace is too long to be passed in directly. Derive the entire approach and summarize it for the critic model, describing decisions made and any possible mistakes in detailed, enumerated steps.

''' {t} '''

### E.7 RUBRIC EXTRACTION (CODING)

You are a reasoning analysis system that (1) identifies errors in models' solutions to GitHub issues and (2) explains how a critic model can identify these same errors in other failed solutions.

You will be given a GitHub issue and the summary of a model's incorrect solution to the GitHub issue. Identify the specific error made by the model that caused it to fail. An error description should be concise (max 25 words), but descriptive enough that it could be easily identified in other models' responses in the future. Do not mention the model in an error: Your description should describe a reasoning mistake in the abstract that can be used by a critic model when evaluating other solutions. For similar reasons, do not make the error overly specific to the problem, as the error should be sufficiently generalizable to other similar issues that may have different details.

You will write instructions in a specific format that will describe how someone could identify the error if it appeared in a different solution for a different issue. Your instructions should be organized into two parts: A "key concept" and "verification details".

* KEY CONCEPT: This is a short keyword or concept that someone can quickly identify as being present or not present in a solution to determine if the error could possibly occur in that issue. This should be the lowest level of granularity: If this keyword or concept is present, then the error cannot possibly be present in the solution. Key concepts should be general: You want to aim for recall, not precision.

* VERIFICATION DETAILS: Provided that one or more of the key concepts are present in a solution, write one or more descriptive explanations for how someone could tell if the error exists in that solution. If one of these descriptions matches up with the contents of a solution, then that means the error has occurred there.

Write your answers in JSON format with your error as a key and your instructions in another dictionary as the corresponding value, i.e. {{⟨error_description⟩: {{"key_concept": ⟨scanning_keyword⟩, "verification": ⟨verification_details⟩}}}}.

Now, here is the data you should use to formulate your output:

GitHub issue: "{tq}"

Summary of the incorrect solution: "{tt}"

Output the error JSON and ***nothing else***:

### E.8 KEYWORD AGGREGATOR (CODING)

A model is trying to match pre-written feedback to different incorrect solutions to GitHub issues. Here are some examples of the pre-written feedback in the corpus:

{samples}

However, there are more possible feedback pieces than can fit in a single prompt. So, the model first labels each issue solution with one or more "key concepts", which it then uses to figure out which feedback might be applicable and should be included in the final prompt.

The following candidate "key concept" labels have been generated. Your job is to consolidate highly similar labels that describe the same concepts, to produce a shorter final list of key concepts.

{keywords}

Group the concepts by the underlying sort of material they describe, and then for each group write a new key concept in the same style that can replace the labels in that group. Organize your output in JSON format with your new labels as keys and the lists of original label indices as the values, e.g., {{"description 1": [1, 15, 62], "description 2": [4, 5, 22, 23, 45], ...}}. Output the JSON and nothing else. Make sure your JSON output has correct Python syntax and all lists have correct closing brackets: Lists should always have an opening bracket "[" as well as a closing bracket "]", e.g., "[4, 55, 120]".

### E.9 RUBRIC KEYWORD APPLICATOR (CODING)

You are a reasoning analysis system that is an expert at tagging summaries of solutions to GitHub issues with labels to accurately cover all relevant aspects of the solution so that it can be searched for in a database later. Given a solution summary and list of keyword labels, identify each label in the list that describes a concept or idea present in the solution. You are aiming for complete coverage - every relevant label should be included.

Go down the list item-by-item to see if each one applies to the solution. Write the name of each label that relates to the solution in a Python list, e.g., ["label1", "label2", "label3", ...]. Output your Python list and nothing else.

All concept labels: '" {r} "'

GitHub issue: '" {q} "'

Solution: '" {t} "'

Matching keywords and concepts (in Python list form):

### E.10 RUBRIC ITEM APPLICATOR (CODING)

You are a critic model that takes summaries of AI solutions to GitHub issues and determines whether or not they will produce a correct solution and why. The quality of the path taken to the final pull request doesn't matter, as long as the final solution submitted is correct and solves the issue.

You have been given an error rubric that identifies specific reasons why a solution may be incorrect based on the model's reasoning process, and provides specific ways to tell if one of these errors has occurred in a solution. Use this rubric to predict whether the solution should be marked "correct" or "incorrect".

First, read through the solution. Then, go through the error rubric, line-by-line, and see if any of the entries and their descriptions describe behavior in the solution that will cause it to fail. To provide transparent documentation of your own reasoning, write an explanation for each rubric item describing why or why not the item explains why the solution will fail or not. Do not skip any items - provide an explanation for each one. Explain one error type per line, and do not abridge the process by saying things like [Continuing through all list items...]. I want all of the items listed, you do not have to ask me again. I do not care how long the output is. Write your reasons in JSON format, e.g., {{1: "explanation for error 1", 2: "explanation for error 2", ...}}.

Then, at the end after your JSON of explanations, you will enter a final database entry using ⟨ANS⟩ brackets. Write a final response of "N/A" if you are confident the GitHub issue solution is correct. Write the error number(s) if the solution will fail, for one or more of the reasons in the error rubric. Write your answer in ⟨ANS⟩ brackets, e.g. ⟨ANS⟩3, 41⟨/ANS⟩, ⟨ANS⟩22⟨/ANS⟩, ⟨ANS⟩N/A⟨/ANS⟩, etc.

Error rubric: {rubric}

GitHub issue: {question}

Solution summary: {trace}

Responses:

### E.11 BASELINE CLASSIFIER (GENERAL)

A reasoning system is given a question to answer, and it outputs a reasoning trace as an intermediate step towards outputting a final answer. Your job is to determine whether the reasoning trace is likely to result in a correct final answer. Respond with "correct" or "incorrect" in ⟨ANS⟩⟨/ANS⟩ brackets, and nothing else.

Question: {q}

Reasoning trace: {t}

### E.12 BASELINE, ALT. 2& 3 (GENERAL)

A reasoning system is given a question to answer, and it outputs a reasoning trace as an intermediate step towards outputting a final answer. Given an excerpt from the outputted reasoning trace, your job is to determine whether the system is likely to result in a correct final answer. Respond with "correct" or "incorrect" in ⟨ANS⟩⟨/ANS⟩ brackets, and nothing else.

Question: {q}

Reasoning trace: {t}

### E.13 BASELINE, ALT. 4 & 5 (GENERAL)

A reasoning system is given a question to answer, and it outputs a reasoning trace as an intermediate step towards outputting a final answer. Given an excerpt from the outputted reasoning trace, your job is to determine whether the system should output an answer now or keep thinking and extend the reasoning trace. Respond with "correct" or "incorrect" in ⟨ANS⟩⟨/ANS⟩ brackets, and nothing else.

Question: {q}

Reasoning trace: {t}

### E.14 BASELINE CLASSIFIER (CODING)

A model is asked to solve a PR request from GitHub, and it outputs a reasoning trace documenting its path towards completing the PR request. Your job is to determine whether the reasoning trace is likely to result in a correct completion of the PR request. Respond with "correct" or "incorrect" in ⟨ANS⟩⟨/ANS⟩ brackets, and nothing else.

Question: {q}

Reasoning trace: {t}

### E.15 BASELINE, ALT. 2& 3 (CODING)

A model is asked to solve a PR request from GitHub, and it outputs a reasoning trace documenting its path towards completing the PR request. Given an excerpt from the outputted reasoning trace, your job is to determine whether the system is likely to result in a correct completion of the PR request. Respond with "correct" or "incorrect" in ⟨ANS⟩⟨/ANS⟩ brackets, and nothing else.

Question: {q}

Reasoning trace: {t}

### E.16 BASELINE, ALT. 4 & 5 (CODING)

A model is asked to solve a PR request from GitHub, and it outputs a reasoning trace documenting its path towards completing the PR request. Given an excerpt from the outputted reasoning trace, your job is to determine whether the system should output an answer now or keep thinking and extend the reasoning trace. Respond with "correct" or "incorrect" in ⟨ANS⟩⟨/ANS⟩ brackets, and nothing else.

Question: {q}

Reasoning trace: {t}

# F    EXAMPLE DATA

## F.1    CHEMICAL ENGINEERING QUESTION

In a mixture to be fed to a continuous distillation column, the mole fraction of phenol is 0.35, o-cresol is 0.15, m-cresol is 0.30 and xylenols is 0.20. A product is required with a mole fraction of phenol of 0.952, o-cresol 0.0474 and m-cresol 0.0006. If the volatility to o-cresol of phenol is 1.26 and of m-cresol is 0.70, estimate how many theoretical plates would be required at total reflux.

## F.2    COMPRESSED CHEMICAL ENGINEERING TRACE

1. Given a multicomponent distillation column operating at total reflux with:
- Feed composition: phenol (0.35), o-cresol (0.15), m-cresol (0.30), xylenols (0.20)
- Desired distillate composition: phenol (0.952), o-cresol (0.0474), m-cresol (0.0006)
- Relative volatilities to o-cresol: phenol (1.26), m-cresol (0.70)

2. Set up material balances for each component:
- Phenol: $0.35F = 0.952D + B_{ph}$
- O-cresol: $0.15F = 0.0474D + B_o$
- M-cresol: $0.30F = 0.0006D + B_m$
Where F is feed rate, D is distillate rate, B is bottoms rate

3. Define $q = D/F$ (distillate fraction) and establish two Fenske equations:
- For phenol/o-cresol: $20.0844 * (0.15 - 0.0474q)/(0.35 - 0.952q) = 1.26^N$
- For phenol/m-cresol: $79.0 * [(0.30 - 0.0006q)/(0.15 - 0.0474q)] = 1.42857^N$

4. Through iterative calculations, found $q \approx 0.265$ where both equations give $N \approx 14.45$:
- With $q = 0.265$:
* $B_{ph} = 0.0977F$
* $B_o = 0.13744F$
* $B_m = 0.29984F$

5. Verified result by checking separation ratios:
- For phenol/o-cresol: $(D_ph/D_o)/(B_{ph}/B_o) \approx 28.26 = 1.26^{14.45}$
- For phenol/m-cresol: $(D_{ph}/D_m)/(B_{ph}/B_m) \approx 4865.5 \approx 1.8^{14.45}$

6. Since number of plates must be an integer and we need to ensure separation, round up to 15 theoretical plates.

### F.3 Math Question

4. Let in the $xOy$ plane, the area enclosed by $0\langle y \leq x^2, 0 \leq x \leq 1$ be $\frac{1}{3}$. Then the intersection of the sets

$$M = \{(x,y) \mid |y| - |x| \leq 1\},$$
$$N = \{(x,y) \mid |y| \geq x^2 + 1\}$$

representing the area of $M \cap N$ is ( ). (A) $\frac{1}{3}$ (B) $\frac{2}{3}$ (C) 1 (D) $\frac{4}{3}$

### F.4 Compressed Math Trace

1. First identify that set $M$ is defined by $|y| \leq |x| + 1$, which forms a region bounded by $y = |x| + 1$ and $y = -(|x| + 1)$

2. Set $N$ is defined by $|y| \geq x^2 + 1$, which forms two regions:
- Above the parabola $y = x^2 + 1$
- Below the parabola $y = -x^2 - 1$

3. The intersection $M \cap N$ requires points satisfying $x^2 + 1 \leq |y| \leq |x| + 1$

4. In the lower half-plane ($y \leq 0$), the intersection contains only discrete points and contributes no area

5. In the upper half-plane ($y \geq 0$), we need points where:
- $x^2 + 1 \leq y \leq |x| + 1$
- This is only possible for $|x| \leq 1$ (since $x^2 + 1\rangle|x| + 1$ when $|x|\rangle 1$)

6. Due to symmetry about the $y$-axis, calculate area in first quadrant and multiply by 2:
- For $0 \leq x \leq 1$, area between $y = x + 1$ and $y = x^2 + 1$
- Vertical distance between curves is $(x + 1) - (x^2 + 1) = x - x^2$

7. Calculate area in first quadrant:
$\int_0^1 (x - x^2)dx = [\frac{1}{2}x^2 - \frac{1}{3}x^3]_0^1 = \frac{1}{2} - \frac{1}{3} = \frac{1}{6}$

8. Total area $= 2 \times \frac{1}{6} = \frac{1}{3}$

Therefore, the area of $M \cap N$ is $\frac{1}{3}$.

### F.5 CODING QUESTION

**caplog.get_records and caplog.clear conflict**

**Description**

`caplog.get_records()` gets decoupled from actual caplog records when `caplog.clear()` is called. As a result, after `caplog.clear()` is called, `caplog.get_records()` is frozen: it does not get cleared, nor does it get new records.

During test set up it is set to the same list as `caplog.records`, but the latter gets replaced rather than cleared in `caplog.clear()`, which diverges the two objects.

**Reproductive example**

```
import logging

def test(caplog) -> None:
    def verify_consistency() -> None:
        assert caplog.get_records("call") == caplog.records

    verify_consistency()
    logging.warning("test")
    verify_consistency()
    caplog.clear()
    verify_consistency()  # fails: assert [<LogRecord: ...y, 8, "test">] == []
```

**Environment details** Arch Linux, Python 3.9.10:

```
Package    Version
---------- -------
attrs      21.4.0
iniconfig  1.1.1
packaging  21.3
pip        22.0.4
pluggy     1.0.0
py         1.11.0
pyparsing  3.0.8
pytest     7.1.1
setuptools 60.10.0
tomli      2.0.1
wheel      0.37.1
```

### F.6 COMPRESSED CODING TRACE

1. Problem Identification:
- The issue was in the 'LogCaptureHandler.reset()' method
- When 'caplog.clear()' was called, it created a new list instead of clearing the existing one
- This broke the reference between 'caplog.get_records()' and 'caplog.records'

2. Investigation Steps:
- Created a reproduction script to simulate the issue
- Verified that creating a new list breaks references
- Tested alternative approaches to clearing the list

3. Solution:
- Changed 'self.records = []' to 'del self.records[:]' in 'LogCaptureHandler.reset()'
- This clears the list in-place while maintaining the original reference
- Verified the fix with a test script

4. Key Decisions:
- Used 'del list[:]' instead of alternatives like 'list.clear()' or 'list[:] = []'
- Kept the change minimal and focused on the core issue
- Maintained backward compatibility
- Did not require changes to test files

5. Potential Issues:
- None identified - the change is straightforward and maintains existing behavior
- The fix is well-tested with both reproduction and verification scripts

6. Validation:
- Created test scripts to verify both the bug and the fix
- Ensured the fix maintains list reference consistency
- Confirmed the solution works with pytest's fixture behavior

The approach was methodical and focused on the root cause, resulting in a minimal but effective fix.

## G   AI USAGE STATEMENT

LLMs were used to format small sections of LaTeX in the paper.

