# OpenReview forum: "Generating Data-Driven Reasoning Rubrics for Domain-Adaptive Reward Modeling"
_ICLR.cc/2026/Conference — Submitted to ICLR 2026_

### Official Review · Reviewer_cC9Q · 2025-10-27

**Soundness:** 3
**Presentation:** 3
**Contribution:** 2
**Rating:** 4
**Confidence:** 4

**Summary:**

This paper proposes a data-driven method to automatically generate granular reasoning rubrics by extracting error patterns from a model's incorrect reasoning traces. These rubrics are then used to enhance an LLM-as-judge's ability to classify unseen reasoning traces, improving the specificity and overall accuracy of an LLM trace correctness classifier. Furthermore, it can serve as a strong reward function for RL.

**Strengths:**

1. The method is clearly presented and easy to follow, with intuitive motivations.

2. Experiment results demonstrate the superior performance of the rubric-augmented method compared to the baseline.

**Weaknesses:**

1. The paper fails to cite or compare against key existing research on rubric-based rewards [1, 2]. This omission makes it difficult to assess the paper's novelty, as it does not clearly differentiate its method from existing rubric-based approaches.

2. The experiments only benchmark the proposed method against a no-rubric baseline. This merely proves that using a rubric is better than not using one. To validate its core contribution, the paper must demonstrate that its specific method is superior to other existing rubric-augmented methods.

[1] Reinforcement Learning with Rubric Anchors
[2] Rubrics as Rewards: Reinforcement Learning Beyond Verifiable Domains

**Questions:**

Please refer to the Weaknesses section above.

---

> ### Author Response · Authors · 2025-12-04
>
> Thank you for your review! We appreciate that you note the approach’s superior performance over baselines and the paper’s clarity. We hope to answer your questions and concerns below:
>
> > The paper fails to cite or compare against key existing research on rubric-based rewards [1, 2]. This omission makes it difficult to assess the paper's novelty, as it does not clearly differentiate its method from existing rubric-based approaches. The experiments only benchmark the proposed method against a no-rubric baseline. This merely proves that using a rubric is better than not using one. To validate its core contribution, the paper must demonstrate that its specific method is superior to other existing rubric-augmented methods.
>
> Thank you for sharing these pieces of relevant work! As these papers were either released on arXiv shortly before the submission deadline ([1]) or submitted to the same conference ([2]), we did not have the resources to reimplement their approaches and run experiments comparing them. Notably, we would like to emphasize that [1] uses human-generated rubrics, which is a very exciting avenue of research, since prior published work has not explored this for technical domains, but fundamentally differs in their primary goal from that of our approach. We specifically aim to use rubrics that are generated entirely autonomously, to produce rubrics that are tailor-made for a specific domain and model of interest, which would be highly costly and generally unscalable if a human rubric creator was necessary. Therefore, we see these cited works as contemporary and also largely complimentary.

---

### Official Review · Reviewer_wdiQ · 2025-10-30

**Soundness:** 2
**Presentation:** 3
**Contribution:** 2
**Rating:** 4
**Confidence:** 3

**Summary:**

This paper presents a data-driven method to address the unreliability of LLM-as-a-judge frameworks for verifying complex, domain-specific reasoning. The core idea is to automatically generate a "rubric," or a granular error taxonomy, by prompting an LLM to analyze incorrect reasoning traces from a given domain. This rubric is then used as a checklist by an LLM-judge to create a more accurate and robust reward function for reinforcement learning. Experiments in coding, math, and chemical engineering show that rubric-augmented LLM judges outperform baseline judges in error detection (particularly specificity) and that RL-trained models using this rubric-based reward can approach the performance of models trained with verifiable "gold standard" rewards, while requiring fewer labels

**Strengths:**

The paper targets a critical bottleneck in training more capable reasoning models: the difficulty and cost of creating reliable reward signals. The proposed solution is intuitive. Instead of asking an LLM to abstractly "grade" a complex trace, the authors use a approach to equip the LLM with a concrete "failure checklist" (the rubric). This reframes an abstract evaluation problem into a more constrained and verifiable classification task.

**Weaknesses:**

The method’s cost-effectiveness remains insufficiently demonstrated. While Figure 6 indicates that the per-step runtime is comparable to the baseline, the baseline itself, relying on a single LLM-judge call already constitutes a major computational bottleneck in RL. A more detailed analysis of token usage and compute overhead between the two-pass rubric judge and the one-pass baseline would be needed to substantiate the claimed efficiency advantage.

The method shows limited generalizability beyond technical domains, performing well primarily in areas with clear, verifiable errors such as mathematics, coding, and chemical engineering. In contrast, the authors’ experiment on “Psychology & Philosophy” (Appendix B.5) reports a Balanced Accuracy of 0.567 using the rubric method, slightly below the baseline of 0.571, highlighting a key limitation in scope. While effective for detecting “catastrophic errors,” the approach falls short as a general solution for enhancing LLM-as-a-judge performance on qualitative, nuanced, or open-ended tasks.

Given that the method is primarily applicable to technical domains such as math and coding—where ground-truth labels are readily available, what is the motivation for employing a llm-based approach in these settings? How does it outperform or complement conventional supervised or verifiable reward training?

**Questions:**

Please see the weakness part

---

> ### Author Response · Authors · 2025-12-04
>
> Thank you for your review! We appreciate that you note the approach’s intuitiveness and how it addresses a key bottleneck in reasoning model training. We hope to answer your questions and concerns below:
>
> > The method’s cost-effectiveness remains insufficiently demonstrated. While Figure 6 indicates that the per-step runtime is comparable to the baseline, the baseline itself, relying on a single LLM-judge call already constitutes a major computational bottleneck in RL. A more detailed analysis of token usage and compute overhead between the two-pass rubric judge and the one-pass baseline would be needed to substantiate the claimed efficiency advantage.
>
> Thank you for identifying this as a point that lacks clarity in the submitted draft. In the paper, we do not intend to make the argument that the rubric approach is more _computationally_ efficient than verifiable rewards, we demonstrate that it is more _label_ efficient. The graph in figure 6 illustrates that the rubric approach is not notably more computationally expensive than a traditional LLM judge. We see that this information was confusing with the figure caption as-is, and so we have updated the figure caption accordingly.
>
> > The method shows limited generalizability beyond technical domains, performing well primarily in areas with clear, verifiable errors such as mathematics, coding, and chemical engineering. In contrast, the authors’ experiment on “Psychology & Philosophy” (Appendix B.5) reports a Balanced Accuracy of 0.567 using the rubric method, slightly below the baseline of 0.571, highlighting a key limitation in scope. While effective for detecting “catastrophic errors,” the approach falls short as a general solution for enhancing LLM-as-a-judge performance on qualitative, nuanced, or open-ended tasks.
>
> Thank you for raising this point, as it was not made clear in the appendix where this result is included: We believe that the lower performance of this experiment was not due to the more qualitative nature of the domain, but due to the fact that the dataset included both psychology and philosophy problems, which (1) encompassed two highly distinct failure mode sets, and (2) has no simple way of obtaining ground truth trace correctness scores without expert trace annotation. We attempted to use LLM judges for this as we did with chemical engineering, but the resulting scores were generally unreliable for this domain. We have elaborated on this distinction in the appendix’s text, as we do not believe it is representative of our method’s ability to apply to qualitative domains.
>
> > Given that the method is primarily applicable to technical domains such as math and coding—where ground-truth labels are readily available, what is the motivation for employing a llm-based approach in these settings? How does it outperform or complement conventional supervised or verifiable reward training?
>
> We would like to clarify our method’s usefulness in highly technical settings (see our point above discussing its application to more qualitative, non-technical domains): While ground truth labels are readily available for existing datasets, collecting ground truth data for novel technical domains is often a highly costly task, sometimes requiring a large number of technical experts annotating data. Mitigating the number of necessary labels has the potential for substantial saved costs when generating new training datasets for novel tasks.

---

### Official Review · Reviewer_eszo · 2025-10-31

**Soundness:** 3
**Presentation:** 3
**Contribution:** 2
**Rating:** 4
**Confidence:** 3

**Summary:**

This paper introduces a method to automatically create "rubrics," or detailed checklists of reasoning errors, by analyzing a model's past mistakes in a specific domain. The goal is to solve the problem of LLMs being unreliable at self-correcting or judging complex reasoning. These data-driven rubrics are then used to build a much stronger LLM-as-judge reward function. Experiments show that using this rubric-based reward function in reinforcement learning can improve a model's task accuracy by up to +45% over standard LLM judges. This method approaches the performance of training with verifiable "gold" rewards but requires as little as 20% of the labeled data.

**Strengths:**

1. This paper addresses a key limitation of LLMs: their difficulty in reliably identifying errors in complex reasoning traces, especially in expert domains (like coding or math) and on problems without simple verifiable answers.

1. The method is shown to be effective. When LLM judges were augmented with the automatically generated rubrics, their ability to correctly identify incorrect reasoning traces (Specificity) improved dramatically—for example, from 12.2% to 63.4% on SWE-Bench and 16.1% to 61.3% on NuminaMath.

1. The rubric-augmented reward function is shown to be highly effective for reinforcement learning. It allows models to achieve task accuracy approaching that of models trained with "gold" verifiable rewards, but while using significantly fewer gold labels (as little as 20% mentioned in the abstract).

**Weaknesses:**

1. A significant limitation highlighted in the appendix is the classifier's poor performance on the training set itself. The authors note that the low specificity scores indicate that the classifier is unable to re-identify these errors it was trained on without the ground truth answers being provided for guidance.

1. The ablation studies show that a larger rubric is not always better. For the coding domain, the smallest rubric size ($n=25$) actually outperformed most of the larger rubrics.

1. The information loss of the "Trace Compression" step is unknown. The compressing LLM might misunderstand the original trace or inadvertently filter out subtle yet critical errors before the rubric extraction stage even begins. A comparison of uncompressed and compressed traces.

1. The paper's primary successes are demonstrated in technical domains with verifiable answers (Math, Coding, Chemical Engineering). However, in the appendix (Section B.5), the method was tested on Psychology and Philosophy problems and performed worse than the baseline classifier. This suggests the approach may not generalize to qualitative domains where errors are more ambiguous or nuanced.

**Questions:**

1. The rubric is generated from the errors of one "imperfect model". How well does this rubric generalize to judging the reasoning traces of other models, especially those with different architectures or training methodologies (e.g., judging a Llama-3 trace with a rubric built from Qwen)?

1. Is it possible for the compressing LLM to mistakenly discard subtle but critical errors, or to misinterpret the final logical path, thereby corrupting the quality of the rubric from the very first step?

---

> ### Author Response · Authors · 2025-12-04
>
> Thank you for your review. We appreciate that you note the approach’s effectiveness for both trace classification and reinforcement learning. We hope to answer your questions and concerns below:
>
> > A significant limitation highlighted in the appendix is the classifier's poor performance on the training set itself. The authors note that the low specificity scores indicate that the classifier is unable to re-identify these errors it was trained on without the ground truth answers being provided for guidance.
>
> Thank you for raising this point: This experiment over the training data shows that we can pick up signals from a set of training data without exactly memorizing the error boundary. While this results in a classifier that has not overfit on the training set, the produced signal is still clearly effective when applied to novel data, as evidenced by our improved classification accuracy on held-out data in Experiment 1.
>
> > The ablation studies show that a larger rubric is not always better. For the coding domain, the smallest rubric size () actually outperformed most of the larger rubrics.
>
> We see this as a positive aspect of the approach. For some domains, smaller rubrics succinctly capture the necessary failure modes that may occur, which reduces overall compute and improves human interpretability.
>
> > The paper's primary successes are demonstrated in technical domains with verifiable answers (Math, Coding, Chemical Engineering). However, in the appendix (Section B.5), the method was tested on Psychology and Philosophy problems and performed worse than the baseline classifier. This suggests the approach may not generalize to qualitative domains where errors are more ambiguous or nuanced.
>
> Thank you for raising this point, as it was not made clear in the appendix where this result is included: We believe that the lower performance of this experiment was not due to the more qualitative nature of the domain, but due to the fact that the dataset included both psychology and philosophy problems, which (1) encompassed two highly distinct failure mode sets, and (2) has no simple way of obtaining ground truth trace correctness scores without expert trace annotation. We attempted to use LLM judges for this as we did with chemical engineering, but the resulting scores were generally unreliable for this domain. We have elaborated on this distinction in the appendix’s text, as we do not believe it is representative of our method’s ability to apply to qualitative domains.
>
> > The rubric is generated from the errors of one "imperfect model". How well does this rubric generalize to judging the reasoning traces of other models, especially those with different architectures or training methodologies (e.g., judging a Llama-3 trace with a rubric built from Qwen)?
>
> Our approach intends to automatically produce model-specific rubrics based on the failure modes that may be produced by a specific model, so the transferability of a rubric between two models depends on the similarity of the models’ behavior. One future direction for this work is to see how rubrics evolve as the traces used as training data are generated by a variable number of models, and explore how model behavioral differences impact the generated rubrics’ quality and adaptability.
>
> > Is it possible for the compressing LLM to mistakenly discard subtle but critical errors, or to misinterpret the final logical path, thereby corrupting the quality of the rubric from the very first step?
>
> While this is possible, our empirical results suggest that this does not happen at a rate that affects the performance improvements we observe over compared methods. The trace compression is a necessary step to reduce what are otherwise extremely long traces in some cases (20K+ tokens) to a manageable length. However, exploring alternative approaches to this step is a very interesting line of future work.

---

### Official Review · Reviewer_wp9h · 2025-11-03

**Soundness:** 2
**Presentation:** 2
**Contribution:** 2
**Rating:** 4
**Confidence:** 2

**Summary:**

This paper proposes an automated approach to construct granular reasoning error taxonomies ("rubrics") to enhance LLM-driven error detection in reasoning traces. The method extracts domain-specific error patterns from incorrect reasoning traces generated by models, organizing them into hierarchical rubrics with keywords for efficient lookup. These rubrics guide LLM-as-judge classifiers in identifying errors in unseen traces. The authors demonstrate that rubric-augmented classifiers improve error identification by up to 11.6% in technical domains (math, coding, chemical engineering).

**Strengths:**

- Novel application of automatic error taxonomy extraction to reasoning trace evaluation
- Multiple domains tested (coding, math, chemistry)
- Potential to reduce annotation costs in specialized domains

**Weaknesses:**

- Rubric generation requires Claude 3.5 Sonnet (closed-source); no experiments with open-source alternatives
- NuminaMath only evaluated on 100/350 validation problems
- How do we know generated rubrics are comprehensive and not redundant?

**Questions:**

- Can you provide the exact prompts used for rubric generation and classification?
- Why use different LLMs for RL training than rubric generation ?
- Do rubrics from one domain transfer to related domains?

---

> ### Author Response · Authors · 2025-12-04
>
> Thank you for your review. We appreciate that you note the approach’s novelty and potential to reduce annotation costs in specialized domains. We hope to answer your questions and concerns below:
>
> > Rubric generation requires Claude 3.5 Sonnet (closed-source); no experiments with open-source alternatives
>
> Thank you for suggesting this improvement to the paper: To show the applicability of our method using open-source models, we re-ran our trace classification experiment (Sec. 5.2) using the open-source Qwen3-235B model for rubric generation as well as classification. We observe significant specificity improvements across all domains (.268 -> .854 for coding, .108 -> .231 for math, and .272 -> .476 for chemical engineering), as well as balanced accuracy improvements on math and chemical engineering (.526 -> .544 and .600 -> .674). The full set of results are now included in Appendix A, Table 3.
>
> > NuminaMath only evaluated on 100/350 validation problems
>
> We have re-run this experiment and updated the result in the experiment table (Table 1). We find that running on the additional problems has a negligible impact on the R1 baseline’s final accuracy (15% -> 15.5%).
>
> > How do we know generated rubrics are comprehensive and not redundant?
>
> The fact that the rubrics generated from training data effectively classify trace failure modes that occur on the validation set demonstrates their comprehensiveness (see Experiment 1). We do not see redundancy as a downside, as our trace classification approach does not inherently degrade in performance if multiple similar failure modes are present in a rubric. If redundancy introduced notable empirical downsides, these would be reflected in experimental results.
>
> > Why use different LLMs for RL training than rubric generation ?
>
> We use a smaller LLM for our RL classifier than for our rubric generation because (1) the rubric generation requires parametric knowledge maintained by large LLMs, while classification makes use of this model’s parametric knowledge as it exists in rubric form and (2) much more compute is necessary for RL training. So, using a smaller model optimizes the performance-cost tradeoff.
>
> > Do rubrics from one domain transfer to related domains?
>
> Our approach intends to automatically produce domain-specific rubrics based on the failure modes that may occur in that specific domain, so there is limited transferability between rubrics produced using trace training sets generated from sufficiently distinct question distributions. We do not see this as a downside, as our goal is to produce rubrics for a specific domain and model that one wants to improve reasoning for.

---

### Author Response · Authors · 2025-12-04

## Summary

Our submission introduced a data-driven approach to automatically construct reasoning error taxonomies to enhance LLM-driven error detection on unseen reasoning traces. We demonstrated that these error taxonomies, or “rubrics”, can be used as an artifact for a trace classifier model to identify traces output by reasoning models that result in incorrect final answers across a range of different domains. Additionally, we demonstrated that such a rubric can be used by a trace classifier as a reward model for training reasoning models to avoid these reasoning failures via reinforcement learning, resulting in improved downstream accuracy over baselines, and approaching verifiable rewards with a fraction of the required gold data labels.

Below, we detail key weaknesses and questions raised by reviewers, along with our responses and/or updates to the paper contents:

### Rubrics’ ability to transfer to other domains/models (wp9h, eszo)

Multiple reviewers asked how the generated rubrics can be used for different models and domains than the model/domain the rubric was generated for. Our approach intends to automatically produce domain- and model-specific rubrics based on the failure modes that may be produced by a specific model in that domain, so the transferability of a rubric between two models or domains depends on the similarity of the models’ behavior and similarity of domain failure modes. One future direction for this work is to see how rubrics evolve as the traces used as training data are generated by a variable number of models, and explore how model behavioral differences impact the generated rubrics’ quality and adaptability.


### Value of rubrics on technical domains (wdiQ)

It was questioned what the point of label efficiency is on technical domains, where existing datasets generally have ground truth labels and verifiable methods for checking scores. We argue that while ground truth labels are readily available for existing datasets, collecting ground truth data for novel technical domains is often a highly costly task, sometimes requiring a large number of technical experts annotating data. Mitigating the number of necessary labels has the potential for substantial saved costs when generating new training datasets for novel tasks.


### Psychology and philosophy experiment results (eszo, wdiQ)

Reviewers expressed concerns regarding the low performance of the rubric approach on the psychology/philosophy domain, which was included as an additional experimental result in the appendix. We believe that the lower performance of this experiment was not due to the more qualitative nature of the domain, but due to the fact that the dataset included both psychology and philosophy problems, which (1) encompassed two highly distinct failure mode sets, and (2) has no simple way of obtaining ground truth trace correctness scores without expert trace annotation. We attempted to use LLM judges for this as we did with chemical engineering, but the resulting scores were generally unreliable for this domain. We have elaborated on this distinction in the appendix’s text, as we do not believe it is representative of our method’s ability to apply to qualitative domains.


### Additional requested experiments (wp9h)

Additional experiments were requested to assess the rubric approach’s efficacy when using an open-source model for rubric construction and trace classification, and to use the full validation set for the R1 baseline. The latter has been run and updated in Sec 5.3, Table 1 (resulting in a small accuracy change of .05). The former experiment has been included in Appendix A, Table 3 using Qwen3-235B: We observe significant specificity improvements across all domains (.268 -> .854 for coding, .108 -> .231 for math, and .272 -> .476 for chemical engineering), as well as balanced accuracy improvements on math and chemical engineering (.526 -> .544 and .600 -> .674).


### Recommended related work (cC9Q)

Contemporary papers were suggested as baselines for the paper. As these papers were either released on arXiv shortly before the submission deadline ([1]) or submitted to the same conference ([2]), we did not have the resources to reimplement their approaches and run experiments comparing them. Notably, we would like to emphasize that [1] uses human-generated rubrics, which is a very exciting avenue of research, since prior published work has not explored this for technical domains, but fundamentally differs in their primary goal from that of our approach. We specifically aim to use rubrics that are generated entirely autonomously, to produce rubrics that are tailor-made for a specific domain and model of interest, which would be highly costly and generally unscalable if a human rubric creator was necessary.

---

### Meta-Review · Area_Chair_ANtt · 2026-01-06

**Summary:**

The reviewers initially converged on a "Marginally Below" rating (all reviewers rated 4). Several common concerns were raised: 1. Reliance on Closed-Source Models; 2. Novelty and Comparisons: Reviewers (especially cC9Q) pointed out a lack of comparison with contemporary rubric-based RL methods, questioning the specific contribution beyond "rubrics are helpful." 3. Generalizability to Qualitative Domains: Experiments in Appendix B.5 (Psychology/Philosophy) showed poor performance, suggesting the method might only work for technical domains with "catastrophic" logical errors; 4.Technical Omissions: Reviewers noted missing exact prompts, small evaluation sets for specific benchmarks (NuminaMath), and potential information loss during "Trace Compression."

**Reviewer Concerns:**

Addressed Concerns:
1. Evaluation Robustness: The authors updated results for the full NuminaMath validation set, showing the R1 baseline remained stable.
3. Notational/Prompting Clarity: The authors committed to including exact prompts and clarified the logic behind using different models for generation vs. RL training (cost-performance optimization).

Outstanding Concerns:
1. Novelty vs. Contemporary Work: While the authors acknowledged new related works, they did not provide a direct head-to-head empirical comparison. Their argument that their method is autonomous (vs. human-generated rubrics) distinguishes the work conceptually, but the empirical superiority over other autonomous rubric methods remains unverified.
2. Qualitative Domain Generalization: The rebuttal explains that the failure in Psychology/Philosophy was due to dataset mixing and lack of ground truth, rather than the qualitative nature of the domain. However, without a successful experimental demonstration in a qualitative field, the concern regarding the method's limited scope (technical only) persists.
3. Information Loss in Compression.

**Reviewer Scores:**

wp9h	4->4/6	Maintain or raise score. Qwen3 results and full benchmark runs addressed core weaknesses.
eszo	4->4/6	Maintain or raise score. Clarified training set performance and qualitative results.
wdiQ	4->4	Score maintained. Clarified label vs. compute efficiency, but generalization concerns remain.
cC9Q	4->4	Score maintained Authors just acknowledged related works.

---

### Decision · Program_Chairs · 2026-01-26

Reject